# Acrolein Induces Retinal Abnormalities of Alzheimer’s Disease in Mice

**DOI:** 10.3390/ijms241713576

**Published:** 2023-09-01

**Authors:** Shuyi Wang, Xiuying Jiang, Weijia Peng, Shuangjian Yang, Rongbiao Pi, Shiyou Zhou

**Affiliations:** 1State Key Laboratory of Ophthalmology, Zhongshan Ophthalmic Center, Sun Yat-sen University, Guangzhou 510060, China; 2Department of Ophthalmology, Affiliated Foshan Hospital, Southern Medical University, Foshan 528000, China; 3School of Pharmaceutical Sciences, Sun Yat-sen University, Guangzhou 510006, China; 4Guangdong Provincial Institute for Vision and Eye Research, Guangzhou 510060, China; 5School of Medicine, Sun Yat-sen University, Shenzhen 528406, China

**Keywords:** Alzheimer’s disease, acrolein, sporadic, retina, retinal vasculature

## Abstract

It is reported that retinal abnormities are related to Alzheimer’s disease (AD) in patients and animal models. However, it is unclear whether the retinal abnormities appear in the mouse model of sporadic Alzheimer’s disease (sAD) induced by acrolein. We investigated the alterations of retinal function and structure, the levels of β-amyloid (Aβ) and phosphorylated Tau (p-Tau) in the retina, and the changes in the retinal vascular system in this mouse model. We demonstrated that the levels of Aβ and p-Tau were increased in the retinas of mice from the acrolein groups. Subsequently, a decreased amplitudes of b-waves in the scotopic and photopic electroretinogram (ERG), decreased thicknesses of the retinal nerve fiber layer (RNFL) in the retina, and slight retinal venous beading were found in the mice induced by acrolein. We propose that sAD mice induced by acrolein showed abnormalities in the retina, which may provide a valuable reference for the study of the retina in sAD.

## 1. Introduction

Alzheimer’s disease (AD) is the most common neurodegenerative dementia with symptoms of memory deficits, and the progressive deterioration of cognitive functions, as well as agitation and aggression. It is estimated that the number of AD patients will grow to 131 million worldwide by 2050, placing a massive burden on individuals and society [1]. According to different pathologies, AD is usually divided into familial AD (fAD) and sporadic AD (sAD) [2]. The common pathologies of sAD and fAD include neurofibrillary tangles (NFTs) formed by excessive phosphorylated Tau (p-Tau) in the cortex and subcortex, the extracellular deposition of β-amyloid (Aβ) forming amyloid plaques, neuronal loss, and brain atrophy [3]. Only 5% of AD cases are fAD, of which the pathomechanism may be based on the amyloid cascade, caused by mutations in the following genes: *APP*, *PSEN1*, and *PSEN2* [4]. Meanwhile, nearly 95% of cases are classified as sAD [4], which is thought to be caused by a combination of genetic and environmental factors, such as diet, hormonal factors, toxicological exposure, and microbiota [2,4,5].

The retina is considered an anatomical and developmental extension of the central nervous system [6,7], and its vascular system also has morphologically and physiologically similar properties to the cerebrum [8]. In addition, the retina can be imaged noninvasively, providing a unique way for the detection of the brain. The most significant change in AD patients is the progressive loss of cognitive function and brain atrophy, but a large number of studies have shown that AD is not limited to the brain but also leads to visual dysfunction, including color discrimination; contrast sensitivity; and visual acuity and integration [9,10,11], suggesting that the central visual pathways are impaired in AD [12]. Retinal dysfunction, alterations of the retinal structure, and optic nerve degeneration were also found in AD patients and animal models [13,14,15]. Moreover, depositions of Aβ [16,17] and over phosphorylated Tau (p-Tau) [18,19,20] were also detected in the retina of AD patients, suggesting that the retina is affected in AD.

The above alterations in the neural visual pathway have been demonstrated in different fAD models such as APP/PS1 mice, 5 × FAD mice, and 3 × Tg-AD mice [21]. However, it is unclear whether sAD causes retinal impairments like in fAD. Recently, we established a novel sAD mouse model induced by acrolein [22]. Acrolein is a highly nucleophilic α-β-unsaturated aldehyde, which not only exists in the environment, but also is an endogenous substance in our bodies [23]. In recent years, studies have shown that acrolein is involved in different pathological processes of AD and plays an important role in AD [24,25]. C57BL/6 mice that were given acrolein (3.0 mg/kg/d, 4–8 weeks) through intragastric administration showed impairments in learning and memory, episodic memory, and olfactory function [22]. At the same time, a variety of AD pathologic features were found in the hippocampus and cortex of mice, including the proliferation of astrocytes and microglia, increased production of β-amyloid protein (Aβ_1-42_), significantly decreased numbers of dendritic spines of neurons in the CA1 and DG regions of the hippocampus, and increased levels of p-Tau^S396/T231^ [22]. In addition, there was also Tau aggregation in the olfactory bulb. These results show that the sAD mouse model induced by acrolein presented typical cognitive and affective disorders and typical AD pathological features, which could simulate the early pathology of clinical sAD comprehensively [22]. Here, this study aimed to investigate the functional and structural changes in the retina of sAD mice induced by acrolein and to investigate whether the related impairments would present AD-like pathologies.

## 2. Results

### 2.1. Acrolein Induced Memory and Learning Impairment in Mice

Learning and memory impairment is a typical characteristic of AD and can be evaluated by the Morris water maze test (MWMT) and step-through test (STT). In the MWMT, the mice were trained over 4 consecutive days to reach the hidden escape platform and then tested on the last day without the platform, recording the latency and the times. In the hidden platform trials, the mice in the control group experienced a learning process, and the escape latency was gradually shortened (Appendix A). The results of the probe test on the last day showed that the number of traveling through the targeted quadrant of mice in the acrolein group of 3 months was significantly less than that of the control group (*p* < 0.01). However, the number of traveling through the targeted quadrant of mice in the acrolein group of 1 month was decreased without significance compared with that of the control group (Appendix A). Meanwhile, it could be seen that the tracks of mice in the control group were concentrated in the quadrant where the platform was located, while the tracks in the acrolein groups of 1 and 3 months were irregularly and dispersedly distributed in each quadrant.

Subsequently, the mice were tested in STT. The fence at the bottom of the darkroom was electrified; when the mice entered the dark compartment from the illuminated compartment, their feet would immediately receive an electric shock and thus, produce a fear memory of the darkroom. A total of 24 h later, the stimulus controller was turned off, and the latency of mice entering the dark compartment from the illuminated compartment and the number entering the dark compartment were recorded within 5 min. Compared with the control group, the error times were significantly increased (*p* < 0.05, *p* < 0.01, respectively), and the latency was significantly decreased in the acrolein groups of 1 and 3 months (*p* < 0.001, *p* < 0.01, respectively) (Appendix A).

### 2.2. Acrolein Induced Apparent Olfactory Impairment in Mice

The buried food pellet test (BFPT) was performed to evaluate the olfactory function of mice. Latency was recorded from the time the mice were placed into the cage to the moment they grasped the food pellet. The results showed that there was no statistical difference in the latency between the acrolein groups and the control group in the first 2 days. On the last day, the latency of mice in the acrolein group of 1 and 3 months were significantly longer than that in the control group (*p* < 0.001, *p* < 0.05) (Appendix A).

### 2.3. Acrolein Induced Tau Accumulation in Mice Retina

The results of Western blot showed that the levels of amyloid precursor protein (APP), Aβ, p-Tau^T231^ and p-Tau^S396^ did not change significantly in the acrolein groups, but the level of Aβ showed an upward trend (Figure 1A–C). Immunohistochemistry (IHC) staining of the mice retinal sections revealed a partly increased p-Tau deposition in the retinas of mice in the acrolein groups than that in the control group (Figure 1D). The deposition of p-Tau was mostly in the retinal nerve fiber layer/ganglion cell layer (RNF/GCL) and inner nuclear layer (INL). In addition, the thioflavin T (ThT)-positive plaques in the retinal sections were significantly increased in the acrolein group of 1 month (*p* < 0.001), 2 months (*p* < 0.01), and 3 months (*p* < 0.001) than that in the control group (Figure 1E). The ThT-positive plaques were mainly displayed in the outer nuclear layer (ONL). The above results indicated that the deposition of Aβ and p-Tau was partly enhanced in the retina of acrolein-induced sAD mice.

### 2.4. Acrolein Induced RNF/GCL Thickness Decrease in Mice

To explore whether acrolein can cause retinal thickness defects, optical coherence tomography (OCT) and hematoxylin and eosin (H&E) staining were used for detection. Compared with the control group, there was a significant decrease in RNF/GCL in the acrolein groups of 2 and 3 months (*p* < 0.05, *p* < 0.01, respectively), but not in the acrolein group of 1 month. In the analysis of total retina (TR) thickness, there was no significant difference between the acrolein groups and control group (Figure 2A,B).

The RNFL and TR thicknesses in the eyes of mice in each group were also detected by H&E staining. Compared to the control group, a significant decrease in the RNFL thickness in the acrolein groups of 2 months and 3 months (*p* < 0.05, *p* < 0.001, respectively) was observed, while no significant difference was found in the acrolein group of 1 month. There was no significant difference in the TR thickness between the acrolein groups and control group (Figure 2C,D).

### 2.5. Acrolein Induced Abnormal Visual Function in Mice

To assess the visual function of mice in acrolein groups and control group, flash electroretinogram (ffERG) and intraocular pressure (IOP) were performed. The implicit time and amplitude of the b-wave of mice were measured in dark-adapted 0.01 ERG. The results showed that there was no significant difference both in the implicit time and amplitude between the acrolein groups and control group, suggesting that acrolein might not affect the rod response in retinas (Figure 3A–C).

In dark-adapted 3.0 ERG, there was no significant difference in the implicit time of the a- and b-wave, or in the amplitude of the a-wave between the acrolein groups and control group. The amplitude of the b-wave was significantly decreased in the acrolein groups of 2 and 3 months (both *p* < 0.05), but not in the acrolein group of 1 month, compared with that in the control group, suggesting that acrolein might affect the rod–cone response in the retina (Figure 3D,F).

In light-adapted 3.0 ERG, there was no significant difference in the implicit time of the a- or b-wave in the acrolein groups compared with that in the control group. The amplitude of the b-wave was significantly decreased in the acrolein group of 2 and 3 months (both *p* < 0.01), but not in acrolein group of 1 month compared with that in the control group, suggesting that acrolein might influence the cone response in the retina (Figure 3G–J). The results of the IOP of the mice showed that there was no significant difference in IOP between the acrolein groups and the control group (Figure 3K). The results above indicate that functional impairment in the retina might be induced by acrolein in this sAD model, rather than IOP.

### 2.6. Acrolein Caused Mild Retinal Vascular Damage

Color fundus photographs of mice revealed a clear optic disc and retinal vessels. In the control group and acrolein groups, the optic nerve heads of the mice were round with a clear boundary. Retinal vessels were radially distributed and the retinas were flat and even. However, it was found that an aperture with a clear boundary appeared in the retina of mice in the acrolein group of 3 months (Figure 4A). To determine the in vivo retinal vascular response, fundus fluorescence angiography (FFA) was used to obtain images at 1, 5, 10, and 15 min after the injection. Retinal vessels were filled quickly, and individual capillaries became distinguishable. Throughout the recording process, no leakage phenomena were found in each group. However, slight retinal venous beading was found in the acrolein group of 3 months (Figure 4B).

## 3. Discussion

AD, a progressive neurodegenerative disease, is the main type of dementia. Only 5% of AD cases are fAD, while nearly 95% are classified as sAD [26]. Currently approved treatments for AD include cholinesterase inhibitors and NMDA-receptor antagonists, and their combination provides only temporary relief [27]. As it is reported, acrolein plays an important role in AD. It might be beneficial for the treatment of AD to scavenge acrolein with drugs such as 2-hydrazino-4,6-dimethylpyrimidine and polyphenols [28].

Acrolein is a highly active toxic aldehyde, a common dietary and environmental pollutant, and can also be produced endogenously. Acrolein is also easily absorbed through the gastrointestinal tract into the systemic circulation [23]. In our previous study, we established a novel sAD mouse model induced by the intragastric administration of acrolein (3.0 mg/kg/d, 4–8 weeks), which showed impairments of learning and memory, episodic memory and olfactory function [22]. At the same time, a variety of AD pathologic features were found in the hippocampus and cortex of mice [22]. These results showed that the sAD mouse model induced by acrolein could simulate the clinical sAD early pathology comprehensively [22]. Therefore, in the present study, a sAD mouse model induced by acrolein was established according to the previous study [22] and used to investigate alterations in the retina for 1, 2, and 3 months. Our study showed that acrolein induced cognitive impairment and olfactory dysfunction in C57BL/6 mice (Appendix A), which was consistent with our previous study and validated this model again [22].

As an extension of the brain, the retina has similar characteristics with the brain, and also provides a unique approach for the study of neurodegenerative diseases. The new concept of the “retina as a potential diagnostic tool for AD” has received extensive attention, and more and more researchers have studied the changes in the retina in AD [21]. Several AD mice models, mainly including transgenic mice with fAD, have been studied regarding their retinas. However, little research has been performed in sAD models. Therefore, it is necessary to investigate whether retinal abnormalities are found in sAD mice, which will provide new insights for the diagnosis and treatment of sAD. Plaques formed by Aβ and tangles formed by p-Tau are specific characteristics of AD. In this study, we found that acrolein induced Aβ increase and p-Tau accumulation in the retina from an early stage (1 month). In Western blot analysis, there was no significant change in the levels of APP, Aβ and p-Tau in the acrolein groups (Figure 1A–C). However, an accumulation of Aβ and p-Tau was shown in IHC and ThT staining in the early stage, suggesting that Aβ and p-Tau were locally deposited in the retina (Figure 1D,E). In the retinas of fAD mice, several studies have reported Aβ plaques in the retina of APP/PS1 mice at the late stage (12–18 months of age) [29,30], while no changes in the level of Aβ were detected in the retina of 3 × Tg-AD mice at the early stage (4–8 months of age), but the level of p-Tau^s396^ was significantly increased at 4 months of age [31]. In addition, Aβ plaques have been reported around various photoreceptors of outer and inner segment of photoreceptor cells and the ONL, as well as the GCL in 9-month-old APP23 mice [32]. The plexus layer of the retina contains dendrites and axons of retinal neurons. It has been reported that Aβ may be transported and released anterograde at the synapse, accumulating as extracellular deposits, similar to the brain of AD mice [33]. With developments in technology, in vivo ophthalmologic detection technology can be also used detect the Aβ deposition in AD patients or animals. Some studies used hyperspectral imaging and found that APP/PS1 mice [34] and 5 × FAD mice [35] showed higher hyperspectral reflectance from 4 months of age, indicating the presence of Aβ deposition. In addition, in vivo imaging of APP/PS1 mice aged 2–3 months using curcumin detected Aβ plaques in mouse retinas (mainly in the GCL and NFL), which was earlier than that in hippocampus and brain tissues [16], suggesting that the detection of Aβ deposition in the retina may be of great significance for the early diagnosis of AD. In this study, although no significant changes were found in the levels of Aβ and p-Tau in acrolein groups, local changes were found in the retinas of sAD mice. This may be related to the modeling time and detection methods, so it is necessary to extend the modeling time or improve detection methods for further study.

The changes in the levels of Aβ and p-Tau in the local retina of mice may cause damage to the cells in the retinal layer, and changes in retinal thickness reflect the changes to each layer and its cells to some extent. Several studies have shown that patients in mild to moderate stages of AD showed a significant reduction in RNFL thickness [11,36]. Therefore, we studied the retinal structure of mice using OCT and H&E staining, and analyzed the thickness changes of the RNFL and TR. We found that the thicknesses of RNFL and TR were significantly reduced in the acrolein groups of 2 and 3 months (Figure 2). Likewise, the thickness of the RNF/GCL in retina was decreased in TgCRND8 mice (4 months) [37]. The RNFL is formed by the axons of RGCs, and damage to RGCs can cause axon atrophy, ultimately leading to a reduction in the thickness of RNFL. In addition, Aβ and p-Tau were mainly accumulated in RNF/GCL, INL and ONL. Therefore, the RNF/GCL thickness may be affected by Aβ and p-Tau accumulation in sAD mice induced by acrolein at an early stage. 

In addition, several reports have demonstrated that functional changes in the retina were found in AD patients, with a decrease in the amplitude and a delay in the implicit time of the pattern electroretinogram (PERG) [38], and a decrease in the amplitude of ffERG [39]. In the present study, dark-adapted and light-adapted ERG were used to assess the retinal changes. The results showed that the amplitudes of the a- and b-wave, and implicit time of dark-adapted 0.01 ERG, were normal in the acrolein groups, suggesting that rod photoreceptors are not affected by acrolein (Figure 3A–C). However, the b-wave amplitude of dark-adapted 3.0 ERG was significantly decreased in the acrolein groups of 2 and 3 months, with no significant changes in the a-wave amplitude and implicit time, suggesting that acrolein has an influence on the combined rod-cone response (Figure 3D–F). The results of the dark-adapted ERGs were partly consistent with those of McAnany et al. [40], who reported that the dark-adapted b-wave amplitude for the 5 × FAD mice was significantly decreased at a flash luminance of 0.01 to 10.0 cd·s·m^−2^. Likewise, the amplitudes of the a- and b- waves in scotopic ERG were significantly decreased in APPswe/PS1ΔE9 mice, with no significant difference in implicit time [29]. However, another study reported that there were no significant changes in the implicit time and the amplitude in scotopic ERG in the App*^NL-G-F^* mice of 3–12 months [41]. The implicit time and amplitudes of the a-wave in light-adapted 3.0 ERG were normal, while the amplitudes of the b-wave were significantly decreased in the acrolein groups of 2 and 3 months (Figure 3H,I). However, the b-wave for flash luminance of 3.0 cd·s·m^−2^ was smaller in the 5 x FAD group compared to the WT in the light-adapted condition [40]. The a-wave is mainly caused by the activity of rod photoreceptors in a dark-adapted condition, and of cone photoreceptors in a light-adapted condition, while a b-wave response is largely generated from the activity of bipolar cells and other interneurons. In addition, no significant difference of IOP was found in the control and acrolein groups (Figure 3K). In conclusion, the bipolar cells and other interneurons may be firstly affected, rather than the rod and cone photoreceptors, in acrolein-induced sAD mice from an early stage, and independent of IOP.

Abnormalities in the blood-retina barrier (BRB) have been reported in AD [42]. In our studies, retinal vascular abnormality was also found in the sAD mice induced by acrolein, evaluated using the color fundus photograph and FFA. The results revealed that no leaks were observed in all groups. In addition, the retinal vascular structure was in normal condition in the control group, acrolein groups of 1 and 2 months. However, slight retinal venous beading was found in the acrolein group of 3 months, suggesting that abnormality appears in the retinal vascular system of acrolein sAD mice at a later stage (Figure 4). Early and progressive reductions in vascular platelet-derived growth factor receptor-β (PDGFRβ) has been reported in patients with mild cognitive impairment, and AD was significantly associated with an increased retinal vascular load of Aβ, leading to BRB abnormalities [42]. The loss and apoptosis of peripheral cells in the retinal vascular system could result in the thinning and dilation of blood vessels, leading to venous beading. Retinal venous beading, a chronic dilation in response to the retinal vein after retinal ischemia or other abnormalities [43], can produce a wide range of clinical manifestations, which includes hard exudates, venous occlusion, retinal neovascularization and leakage [44].

In summary, our study showed that an increase in Aβ and p-Tau were partly observed in the retina of sAD mice induced by acrolein. In addition, retinal structure and function impairments were also found at the early stage, while retinal vascular abnormality was found at the later stage, suggesting that retinal vascular impairment might happen subsequently in sAD mice. This is the first time that alterations in the retina of the sAD model induced by acrolein has been reported, which could mimic clinical cases. Further research should be performed to better understand the changes in the retina and even other aspects of the eyes in sAD, which will also provide new insights for the diagnosis and treatment of AD.

## 4. Materials and Methods

### 4.1. Experimental Model and Subject Details

Male C57BL/6 mice of 7 to 8 weeks old were purchased from Beijing Huafukang Experimental Animal Technology Co., Ltd. (Beijing, China) The mice were raised in standard laboratory conditions (12 h light/dark cycle) with adequate food and water. Acrolein was freshly dissolved in ultrapure water. The animals were gavage-fed with acrolein (3.0 mg/kg/d) or ultrapure water of the same volume daily for 1, 2 and 3 months, respectively.

### 4.2. STT

The STT was performed according to the method described previously [45] in a shuttle-box, which was divided into an illuminated compartment and a dark compartment, and the floor of the shuttle-box was made of metal rods. The test lasted for 3 days. On the first day, the mice were allowed to move freely in the shuttle-box for 5 min to adapt the experimental environment. On the second day, the mice were placed in the illuminated compartment. When a mouse entered the dark compartment, it received an inescapable foot shock (50 Hz) delivered through the metal floor. On the last day, the mice were placed in the illuminated compartment, and no electric foot shock was applied in the dark compartment. The times of entering the dark compartment from the illuminated compartment and the latency were recorded over 300 s.

### 4.3. MWMT

The individual learning and memory of the animals were evaluated using MWMT as described previously [46]. The test was performed in a circular tank, which was filled with water (25 ± 2 °C). The pool was divided into four equal quadrants labeled different signs with an escape platform remaining in the same quadrant of the pool. Before training, the mice were allowed to swim freely and reach the visible platform in 60 s. During day 1–4 training, the platform was hidden by adjusting it 1 cm below the surface of the water. The mice were put into the water from four quadrants, respectively, and allowed to reach the escape platform freely in 60 s. After climbing on the platform, the mice remained there for 20 s. If the mice were not able to reach the platform, they were guided to the platform and allowed to remain there for 30 s. All mice were trialed in each quadrant for five days continuously. During the training test, the latency period to reach the escape platform was measured. A probe test was performed 24 h after the last day of training. The escape platform was removed. The mice were put into the water and allowed to swim in the pool without the escape platform for 60 s. The latency and times of crossing the position where the platform used to be were recorded using an automated tracking system.

### 4.4. BFPT

The BFPT was performed with modifications [47]. Before the experiment, the mice were restricted to 0.2 g of food for 18–24 h, but allowed free access to water. For the subsequent 3 days, they underwent behavioral trials. The test cage was covered with 3 cm bedding material. The mice were placed gently into the cage, in which a food pellet was buried 0.5 cm below the surface of the bedding material. The location of the food pellet was randomly changed each day. The latency was recorded when the mice grasped the food pellet with their forepaws or teeth. The mice were allowed to find the pellet within 5 min and were then returned back to their home cage.

### 4.5. Western Blotting Analysis

Western blotting analysis was performed as previously described [48,49]. In brief, whole retinas were isolated and homogenized in ice-cold lysis buffer, supplemented with protease and phosphatase inhibitor cocktail (Shanghai Beyotime Biotechnology Co., Ltd., Shanghai, China). The protein concentration was detected using a BCA assay kit (Beyotime). Equal amounts of protein (20 μg) were loaded in each lane, separated by 10% SDS-PAGE and electrically transferred to a polyvinylidene fluoride membrane. After blocking in 5% tris-buffered saline (TBS) containing 0.1% Tween-20 and 5% skimmed milk, the membranes were incubated with primary antibodies for APP (1:10,000, ab32136, Abcam, Cambridge, MA, USA), Aβ_42_ (1:10,000, ab2539, Abcam), phospho-tau S231 (1:3000, ab151559, Abcam), phospho-tau S396 (1:10,000, ab109390, Abcam) or β-actin (1:1000, 4970S, Cell Signaling Technology, Inc., Danvers, MA, USA) overnight at 4 °C. After incubation with the horseradish peroxidase-conjugated secondary antibodies, the protein bands were visualized by using a BeyoECL Plus kit (Beyotime). The intensity of bands was analyzed using ImageJ software (http://imagej.nih.gov/ij/).

### 4.6. Histological Examinations

The mice were anesthetized an overdose of 1% pentobarbitone solution. Immediately after euthanasia by luxation of the cervical spine, the eyes were excised and fixed in 2% glutaraldehyde and 2% paraformaldehyde solution (pH 7.4) for 24 h at 4 °C.

The eyes were embedded in paraffin, and then 4 μm thick sagittal sections of the eyes were made using a rotary microtome (Leica Microsystems, Wetzlar, Germany). Eye tissues were mounted and subjected to HE staining (Wuhan Servicebio Technology CO., Ltd., Wuhan, China). The thicknesses of RNF/GCL and TR were evaluated and analyzed. The images were acquired and analyzed using a fluorescence microscope (Eclipse Ci-S; Nikon, Tokyo, Japan) and ImageJ software.

### 4.7. IHC

The sections were deparaffinized in gradient-concentration xylene and ethyl alcohol. For IHC, after repairing by heating in pH6.0 citrate antigen retrieval solution (Beyotime), endogenous peroxidase activity was suppressed with 3% H_2_O_2_ for 10 min. The sections were blocked with 20% goat serum blocking solution at room temperature for 60 min, and incubated overnight at 4 °C with the following primary antibody: phospho-tau S181 (1:200, 12885S, CST) or phospho-tau S231 (1:500, Abcam). After 3 washes with PBS, the sections were incubated with horseradish peroxidase conjugated goat anti-rabbit IgG (1:500, G-21234, Invitrogen, Thermo Fisher Scientific, Inc., Rockford, IL, USA) for 1 h at room temperature. The sections were developed with diaminobenzidine tetrahydrochloride (Beyotime) and were counterstained with hematoxylin. The images were acquired and analyzed using the Nikon fluorescence microscope.

For ThT staining, the sections were incubated with 0.1% ThT (Shanghai Macklin Biochemical Co., Ltd., Shanghai, China) for 10 min at room temperature, followed by removal of excess ThT with 70% alcohol. The sections were stained using TBS/glycerol mounting medium. The images were acquired and analyzed using the Nikon fluorescence microscope and imageJ software.

### 4.8. OCT Imaging

For OCT imaging, the mice were anesthetized using an intraperitoneal injection of 1% pentobarbitone solution at 50 mg/kg. Both pupils were dilated with eye drops containing a mixture of 0.5% tropicamide and 0.5% phenylephrine hydrochloride (the pharmacy of Zhongshan Ophthalmic Center, Guangzhou, China). Then, 1% hydroxypropyl methyl cellulose was applied to the eyes of mice to preserve corneal hydration. The mice were placed on a custom stage that allowed for free rotation to acquire volumetric images of centered on the ONH by using SD-OCT (Envisu 2300, Leica Microsystems, Wetzlar, Germany). All images were acquired within 20 min of the onset of anesthesia to avoid the loss of signal intensity due to the occurrence of reversible cataracts [50]. TR and RNF/GCL thickness were measured and analyzed. All mice were scanned in a round area centered on the optic disc.

### 4.9. IOP Measure

The mice were gently grabbed and immobilized to open their eyes without anesthesia, and then IOP was non-invasively measured using a handheld electronic tonometer (Icare Tonolab, Helsinki, Finland). Three consecutive readings were obtained from each eye and the average was regarded as the actual IOP.

### 4.10. ERG Recording

Dark- and light-adapted ffERG were recorded successively using a RETI-port system (Roland Consult, Brandenburg, Germany) according to the ISCEV standard as preciously described [41,51]. Prior to testing, the mice were dark-adapted overnight, then anesthetized using an intraperitoneal injection of 1% pentobarbitone solution at 50 mg/kg. Then, 1% hydroxypropyl methyl cellulose was applied to the eyes of mice to preserve corneal hydration after pupil dilation with eye drops containing a mixture of 0.5% tropicamide and 0.5% phenylephrine hydrochloride. A loop electrode was placed on the cornea as a positive electrode, a needle reference electrode and a needle ground electrode were placed subcutaneously in the lateral cantus and tail base, respectively. First, dark-adapted ERGs were recorded at flash intensities of 0.01 and 3.0 cd·s·m^−2^. For each intensity, three flashes were averaged with data collecting within 300 ms. The bandpass filter was 75–300 Hz. Dark-adapted 0.01 ERG (rod response) reflects the function of the rod system, which occurs in the retinal rod bipolar cells and is the only standard test that selectively monitors rod system function [52]. Abnormality of the dark-adapted 0.01 ERG can be caused by either rod photoreceptor dysfunction or selective dysfunction occurring post-phototransduction or at the level of the rod bipolar cells [52]. Dark-adapted 3.0 ERG (combined rod-cone response) reflects both rod and cone systems, but the contribution of rod system is dominant in the normal retina [52].

Light-adapted ERG was recorded at flash intensities of 3.0 cd·s·m^−2^ after light adaption, with same parameters as dark-adapted ERG. Light-adapted 3.0 ERG (cone response) reflects the function of cone system. Abnormality of the light-adapted 3.0 ERG can be caused by either cone photoreceptor dysfunction or selective dysfunction occurring post-phototransduction or at the level of the cone bipolar cells [52].

Analysis was performed with pCLAMP Software version 11 (Molecular Devices, LLC., Sunnyvale, CA, USA). The implicit time and amplitude of the a- and b-waves for ERG were measured.

### 4.11. FFA

The images of the mice ocular fundi were obtained using a fundus camera (PHOENIX Micron IV, Phoenix Research Laboratories Inc., Pleasanton, CA, USA). Following anesthetization, pupil dilation, and the application of 1% hydroxypropyl methyl cellulose, the mice were placed on the operating stage. Fundus images were taken under white light. Then, the mice were intraperitoneally injected with 0.25 mL 2.5% sodium fluorescein (Macklin). Photographs were taken at 1–15 min post-injection.

### 4.12. Quantification and Statistical Analysis

Statistical analyses were performed using Student’s *t*-test through the program GraphPad Prism 8.0 software (GraphPad Software Inc., La Jolla, CA, USA). Data are presented as mean ± SEM. The significance value was set at *p* < 0.05 (* *p* < 0.05, ** *p* < 0.01, and *** *p* < 0.001).

## 5. Conclusions

In conclusion, alterations in retinal function and structure, levels of Aβ and p-Tau in the retina, and the retinal vascular system were found in this sAD mouse model induced by acrolein, which will provide new insights for the diagnosis and treatment of sAD.

## Figures and Tables

**Figure 1 ijms-24-13576-f001:**
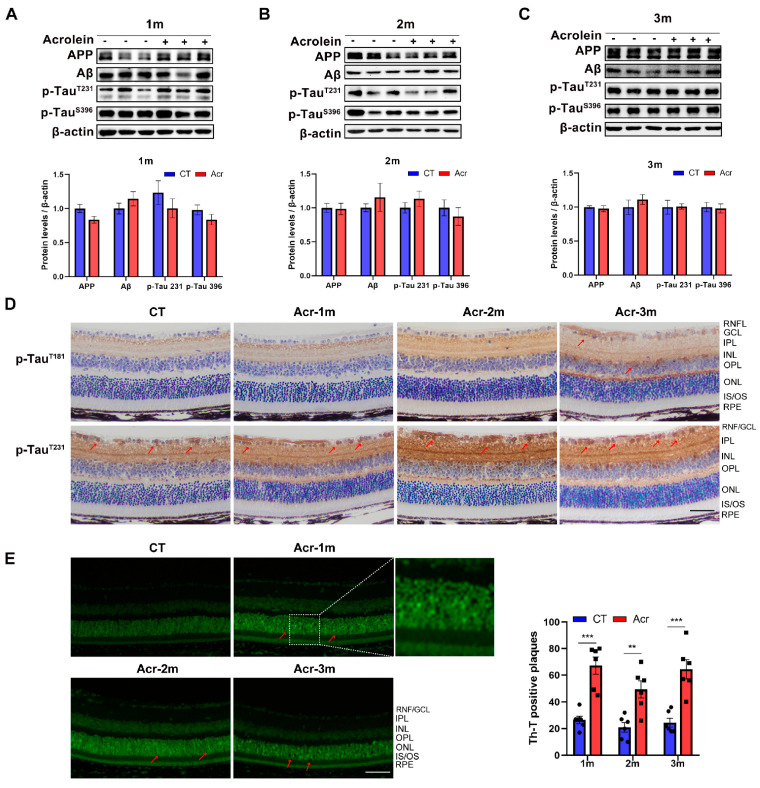
Acrolein partly induced tau accumulation in mice retina. (**A**–**C**) Western blot and quantification of the levels of amyloid precursor protein (APP), β-amyloid (Aβ) and phosphorylated Tau (p-Tau) in the retina in the control group and the acrolein groups. (**D**) Representative immunohistochemistry (IHC) staining for p-Tau^T181^ or p-Tau^T231^ antibodies (red arrow) in the retinas. (**E**) Representative staining and analysis of thioflavin T (ThT)-positive plaques (red arrow) in the retinas of mice. *n* = 4–7 for Western blot, *n* = 6 for IHC and ThT staining. Student’s *t*-test was used. Data are presented as mean ± SEM. ** *p* < 0.01, *** *p* < 0.001 vs. control. Abbreviations: RNFL = retinal nerve fiber layer, GCL = ganglion cell layer, IPL = inner plexiform layer, INL = inner nuclear layer, OPL = outer plexiform layer, ONL = outer nuclear layer, IS/OS = photoreceptor inner and outer segment, RPE = retinal pigment epithelium. Scale bar: 50 μm.

**Figure 2 ijms-24-13576-f002:**
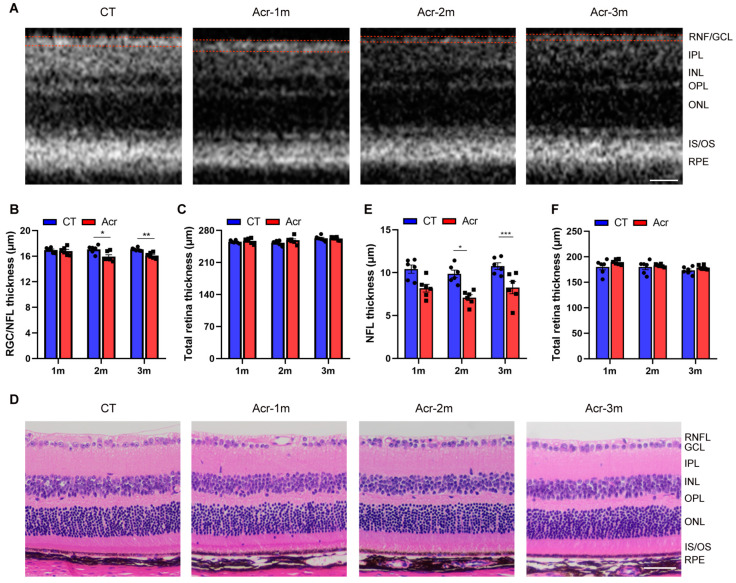
Acrolein induced decreased thickness of retinal nerve fiber layer/ganglion cell layer (RNF/GCL). (**A**) Representative pictures of mice in each group based on optical coherence tomography scans. (**B**,**C**) Analysis of RNF/GCL and total retina thickness. (**D**) Representative pictures of mice in each group based on hematoxylin and eosin staining. (**E**,**F**) Analysis of RNFL and total retina thickness. *n* = 6 for each group. Student’s *t*-test was used. Data are presented as mean ± SEM. * *p* < 0.05, ** *p* < 0.01, *** *p* < 0.001 vs. control. Scale bars: 50 μm.

**Figure 3 ijms-24-13576-f003:**
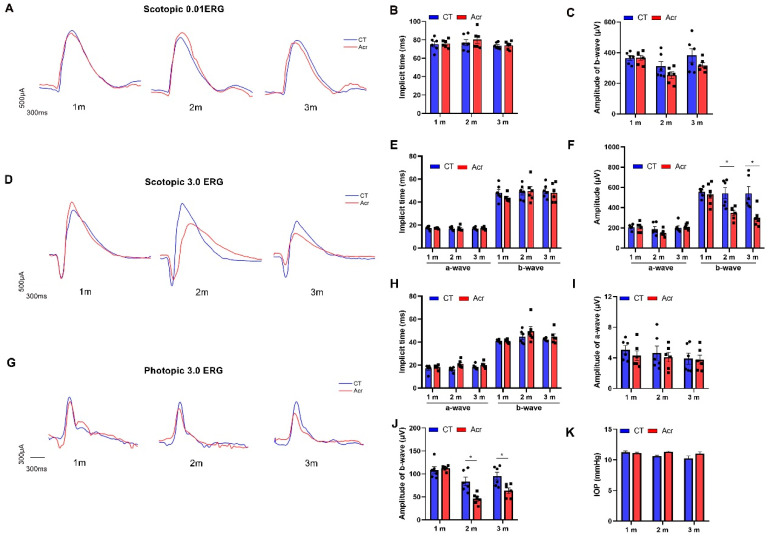
Acrolein induced abnormal visual function in mice. (**A**) Representative traces, and (**B**,**C**) analysis of implicit time and amplitude of the b-wave in scotopic 0.01 electroretinogram (ERG). (**D**) Representative traces of scotopic 3.0 ERG, and (**E**,**F**) analysis of implicit time and amplitude of a- or b-waves in scotopic 3.0 ERG. (**G**) Representative traces, and (**H**–**J**) analysis of implicit time and amplitude of a- or b-wave in photopic 3.0 ERG. (**K**) Mean intraocular pressure (IOP) of mice. *n* = 6 for each group. Student’s *t*-test was used. Data are presented as mean ± SEM. * *p* < 0.05 vs. control.

**Figure 4 ijms-24-13576-f004:**
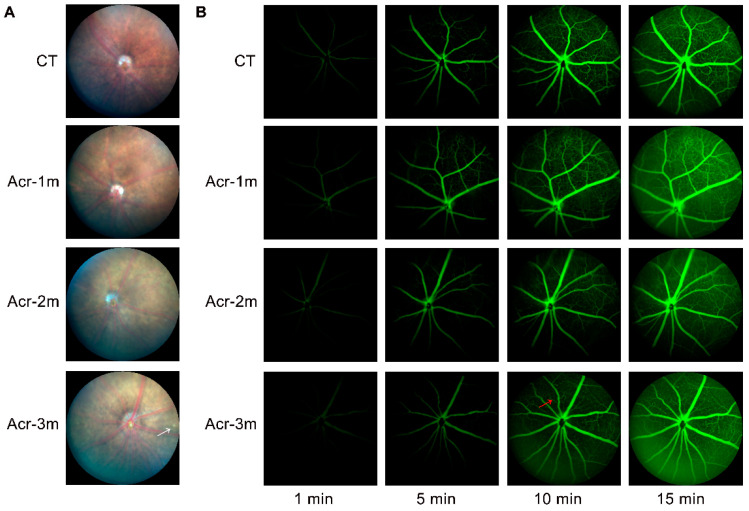
Acrolein induces retinal vascular abnormality. (**A**) Representative images of fundus photography for mice. An aperture with clear boundary appeared in the retina of mice in acrolein group of 3 months (white arrow). (**B**) Representative images of fluorescence angiography for mice. Slight retinal venous beading was found in acrolein group of 3 months (red arrow). *n* = 6 for each group.

## Data Availability

Not applicable.

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
