# Peer review of "Acrolein Induces Retinal Abnormalities of Alzheimer’s Disease in Mice"

_ijms, 2023, doi:10.3390/ijms241713576_

Round 1

Reviewer 1 Report

see attached file

Reviewer 2 Report

The Authors use a new model of Alzheimer disease recently introduced to mimic the sporadic cases.  Here, the goal is to study the effect  acrolein treatment has on the retina. In agreement with behavioral tests demonstrating Alzheimer's typical impairments, the Authors found an increased accumulation of p-tau and Aβ accompanied by a decreased thickness  of retinal nerve fiber layer (RNFL) and slight retinal venous. These molecular and anatomical changes had a physiological relevance because the acrolein-treated mice showed decreased amplitude of b-wave in scotopic and photopic electroretinogram. The paper is well written and the results well explained. I would like to suggest to add a small comment on why acreolin has been choosen to create this model.

Reviewer 3 Report

Alzheimer’s disease (AD) is a progressive neurodegenerative disease of the brain that affects the patient’s cognitive function and hampers the patient’s ability to perform day-to-day tasks. It is the leading cause of dementia and with increasing number of AD cases throughout the US , the need for understanding the molecular basis of the disease to find therapeutic solutions is of great importance. AD is attributed due to mutations in the pathogenic genes that have been identified to be cause of  familial AD (fAD). AD can also occur sporadically as an effect of environmental and toxicological exposure and this form of AD is referred to as sporadic AD (sAD). While a lot of study exists for fAD, sAD remains unexplored. Moreover, several published literature have shown a link between AD pathogenesis and visual function impairment.  Wang et al., in their manuscript and trying to study this link using an acrolein-induced sAD mouse model.

The significance of this manuscript lies in the study of this link between AD and retinal dysfunction using this unique acrolein-induced sAD mouse model.

The scientific premise of this manuscript is sound and valuable in the field of AD diagnostics. However, the evidence supporting their idea is inadequate.  Here are my following comments:

1] Figures 1 and 2 and confirmation of their earlier published paper and can be moved to supplemental section.  

2a] Fig3E, the figure legend mentions red arrows that show the presence of the plaques, but no red arrows are present on the figure. Providing high magnification images of the thioflavine T IHC panel will help readers understand the presence of these plagues. Please make sure that the figures numbers used in the text to explain your observation match; line 117, I believe it should be Fig3D.

2b] Figures 3F and G are missing or mislabeled.

2c] Can you comment on the difference between p-Tau T181 versus pTau-T231 in AD?  From the images provided, it seems like the intensity p-TauT231 levels in 3m acrolein treated retinas are reduced compared to 2m treated. Can you please comment on it?

3] Figure4: Can you draw a representative boundary along the area that you are measuring? The images are blurry and hard to understand what is being measured.

4a] Fig5: The widely accepted acronym for performing standard ISCEV protocol is called ffERG or simply ERG. The using of FERG term is slightly confusing.

4b] The representative traces shown for cone responses (photopic 3.0 ERG) show that the b-wave amplitude at 3m months is slightly higher than at 2m, almost like a partial rescue of the 2m phenotype is ocurring. Is that possible, can you comment on it?

4c] Have you studied if the ERG changes are associated with any morphological changes in the photoreceptor structure in these mice? Are the photoreceptors degenerating?

5] In the last paragraph, lines 313-321, the authors suggest retinal structure and functional impairments were partly observed in acrolein-induced sAD mice. But from the data in the paper, these changes do not seem to be too drastic. As such, aging the mice longer on acrolein treatment and then studying the morphology of retinal structure might be more useful.

6] With an almost intact BRB (except for one spot of venous beading), how is acrolein given through an oral gavage creating this retinal effect? A comment regarding this will be good to include in the discussion section.

There are only some minor changes in the language required. Overall, the language is not hard to follow.

Round 2

Reviewer 3 Report

The authors have made the necessary modifications and I have no further comments.